# ACTIONFILLER: FILL-IN-THE-BLANK PROMPTING FOR OS AGENTS

## ABSTRACT

Many existing methods for operating system (OS) agents focus on predicting the next action based on the current state, which constructs a predefined task execution pipeline. While these methods demonstrate promising performance, reliance on state cognition modules like detector or recognizer could impede execution efficiency, particularly in long-horizon tasks with intricate action trajectories. Recognizing the remarkable accuracy of large language models (LLMs) in processing short instructions, this paper proposes the **ActionFiller** framework. The goal is to integrate easily executable short tasks into longer, cohesive tasks using fill-in-the-blank prompts, thereby minimizing redundant operations and enhancing efficiency. ActionFiller employs two types of action-oriented fill-in-the-blank prompts: one designed for subtasks and another for specific actions. To generate subtask prompts, we introduce a Foresight Optimization Agent (FOA) that constructs an initial prompt by referencing past short tasks. It then fills in the unreferenced parts with detailed prompts generated by a planning agent, effectively retaining valuable past experiences. Next, an Action Template Agent (ATA) generates action prompts for each subtask. This process yields three distinct types of action prompts: 1) executable action sequences, 2) non-executable action sequences with prompt parameters, and 3) pure text descriptions. To execute the action prompts effectively, we propose the CohesiveFlow method, which optimizes the second and third types of prompts by leveraging the cognitive state of the environment. Inspired by masked language modeling, the CohesiveFlow agent integrates the current environmental state with previously executed action sequences to update parameters and text descriptions, ensuring both feasibility and effectiveness in execution. To validate the efficacy of our approach for long-horizon instructions, we introduce a new benchmark called **EnduroSeq** and conduct experiments using the WinBench short instruction dataset. The results demonstrate that ActionFiller significantly enhances task completion rates and execution efficiency, offering a novel solution for the application of intelligent agents in complex environments.

## 1 INTRODUCTION

The development of language models (LM) has led to the emergence of AI-based agentsWang et al. (2024b); Xi et al. (2023), which play diverse roles in facilitating planning, decision-making, and reflection in both single-agentGe et al. (2024); Wang et al. (2023b) and multi-agentHong et al. (2023); Wu et al. (2023) scenarios across various instructionsGe et al. (2023). Currently, operating system (OS) agentsZhang et al. (2024); Humphreys et al. (2022); Hong et al. (2024); Gur et al. (2023); Wang et al. (2024a) primarily rely on two methodologies: constructing execution pipelines for predefined tasksWang et al. (2024a) or using trained models to predict actions based on the current stateZhang et al. (2024); Hong et al. (2024).

In an OS, agents analyze the current state to predict subsequent actions through a decision-making mechanism that evaluates this state and selects the most optimal action. However, before making decisions, state cognition modules—such as icon and text detectors—are employed to assess the running environmentHong et al. (2024). This approach can reduce execution efficiency and prolong execution times, especially with complex long instructions. Moreover, this decision-making paradigm differs significantly from human cognitive processes. Humans tend to optimize their choices based

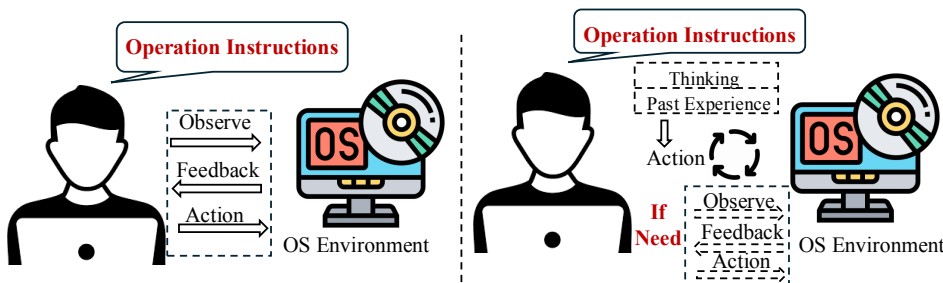

Figure 1: Comparative analysis of general OS agents and our ActionFiller.

on past experiences, often evaluating processes more streamlined before execution. This discrepancy prompts a reevaluation of the methodologies used by OS agents in search of more effective solutions. To further highlight these differences, we have illustrated this distinction in Figure 1 that clarifies the approaches of OS agents versus Our ActionFiller.

To address these challenges, this paper introduces the ActionFiller framework—a novel approach designed to efficiently generate action sequences for operating system (OS) agents. The primary goal is to integrate easily executable short tasks into longer, cohesive tasks using fill-in-the-blank prompts, thereby minimizing redundant operations and enhancing overall efficiency. Unlike traditional methodsSignificant Gravitas; Hong et al. (2023); Zhang et al. (2024) , which rely heavily on cognitive decision-making, ActionFiller provides a more flexible solution by automating the creation of specific action templates and simulating human-like decision-making processes. This enhancement improves the responsiveness and adaptability of OS agents.

The ActionFiller framework consists of two types of action-oriented prompts: one for subtasks and another for action sequences. Subtask prompt: The objective of the subtask prompt is to construct a coherent sequence of steps that reflects human experience while balancing effectiveness and operational flexibility. To achieve this, we introduce a **Foresight Optimization Module** (FOM). Initially, this module references past human experiences to generate a prompt that incorporates both reference steps and additional operational steps. Subsequently, a more detailed prompt is created, devoid of human experiential references, outlining potential subtasks. Finally, this second prompt optimizes uncertain aspects by integrating both human experience and operational flexibility. Action prompt: an **Action Template Agent** (ATA) use the subtask prompt to generates three distinct types of action prompts for execution: 1) Executable Action Sequences: These templates are derived from human experiences and contain tasks with short instructions. This memory can be populated by the language model (LM) using predefined templates or by human input based on specific contexts. Executable action sequences can be directly executed by the OS agent. 2) Unexecutable Action Sequences: Characterized by variable parameters, these sequences cannot be executed without additional context or information. Once the parameters are updated based on the current environmental state, they can become executable. 3) Pure Textual Descriptions: This type emphasizes conveying actions through natural language, providing a narrative or illustrative format. However, these descriptions often exceed the LM's immediate capabilities, necessitating further elaboration or context for effective execution.

To address the limitations of the latter two types of prompts that cannot be executed directly, we propose the **CohesiveFlow** method. Inspired by masked language modeling, the CohesiveFlow agent integrates current environmental data and executed action sequences to refine parameters and decompose textual descriptions into actionable sequences, ensuring both feasibility and effectiveness.

To evaluate the efficacy of ActionFiller, we introduce a benchmark called **EnduroSeq**, specifically designed to assess long-horizon instruction execution. Complementary experiments were conducted using the WinBench short instruction dataset. Experimental results indicate that ActionFiller significantly enhances task completion rates and execution efficiency, offering a transformative solution for the deployment of intelligent agents in complex environments. In summary, our contributions can be summarized as follows:

- We focus on the often-overlooked issue of decision efficiency and propose a novel framework termed ActionFiller to streamline the generation of action sequences. This framework

reduces reliance on cognitive decision-making processes, improves the utilization of memory packages, and enhances execution efficiency for operating system agents.

- To optimize action templates, we introduce the CohesiveFlow method, which optimizes unexecutable action sequences by dynamically updating parameters and leveraging environmental contexts, thereby facilitating more effective decision-making.

- We also present the EnduroSeq benchmark, specifically designed to evaluate long-horizon instruction execution, providing comprehensive validation of our approach.

- Our experimental findings demonstrate that ActionFiller not only increases task completion rates but also improves the adaptability of agents in diverse and complex scenarios, paving the way for more responsive AI-driven solutions.

## 2 RELATED WORK

### 2.1 LLM-BASED OS AGENTS

Yao et al. (2022) and Deng et al. (2024) improved agent performance in real web tasks by developing high-quality web task datasets. Gur et al. (2023) automated the processing of these tasks through the use of pre-trained language models (LLMs) and self-experience learning, while Zheng et al. (2024a) utilized GPT-4V for visual comprehension and web operations. As for the user interface (GUI), Wang et al. (2023a) transform graphical information into HTML representations, incorporating application-specific domain knowledge with LLMs. Yan et al. (2023) introduced a multimodal intelligent mobile agent utilizing GPT-4V, investigating its ability to interpret annotated screenshots. Zhang et al. (2024) replicated human spatial autonomy in managing mobile applications by utilizing XML files for localization, while Wang et al. (2024c) employed visualization module tools for the same purpose, thereby removing the dependency on XML files. Moreover, Hong et al. (2024) created a GUI agent founded on pre-trained visual language models. Zhang et al. (2024) developed a UI multi-agent framework specifically designed for the Windows operating system.

Although various text and visual language agent models have undergone extensive testing across web, mobile, and desktop environments—including UFO Zhang et al. (2024), CC-Net Humphreys et al. (2022), AiTW Rawles et al. (2024), CogAgent Hong et al. (2024), MM-Navigator Yan et al. (2023), SeeAct Zheng et al. (2024b), WebAgent Gur et al. (2023), OS-Copilot Wu et al. (2024), and MobileAgentWang et al. (2024a)—the effectiveness of task reuse, especially in handling complex instructions, still necessitates further investigation.

### 2.2 LARGE MULTIMODAL MODELS

In recent years, Large Multimodal Models (LMMs) have made significant progress, particularly GPT-4V OpenAI (2023) and Gemini Team et al. (2023). Several studies Akter et al. (2023); OpenAI (2023); Yang et al. (2023b); Zhang et al. (2023); Yang et al. (2023a); Yan et al. (2023) highlight their exceptional integration in visual and linguistic reasoning capabilities, demonstrating powerful multimodal skills.

Although open-source models perform well on certain benchmark tests, there is still a performance gap compared to GPT-4V. However, these open-source models have advantages in terms of controllability and ease of fine-tuning, making them suitable for various applications. For example, CogAgent Hong et al. (2024) has been fine-tuned on HTML and screenshot pairs to enhance web understanding capabilities and has improved the processing of high-resolution image details through an image encoder. Additionally, Ferret You et al. (2023) can provide visual referencing and localization functionalities after fine-tuning. These models have also had their capabilities in visual and linguistic understanding and reasoning confirmed by further research from Kazemzadeh et al. (2014); Goyal et al. (2017); Hendrycks et al. (2020); Saikh et al. (2022); Lu et al. (2022); Zhong et al. (2023); Yue et al. (2024).

## 3 ACTIONFILLER

In this section, we first give the pipeline of ActionFiller in subsection 3.1, then provide the generation process of two types of fill-in-the-blank as well as the action execution process.

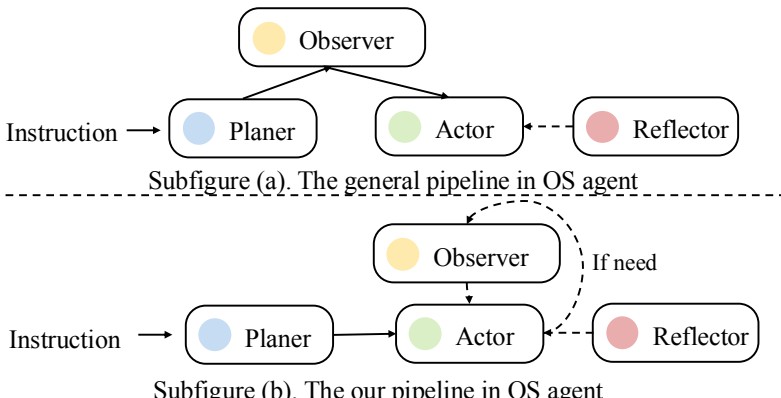

Figure 2: Comparative illustration of general OS agent and ActionFiller in Pipelines.

## 3.1 PIPELINE DEFINITION

In this paper, we consider an agent that employs a large language model, denoted as $\mathcal{L}$, in conjunction with a text-based memory, $\mathbf{M}$, using the Windows operating system as an example. To address the instructions provided by a human user, represented as $q$, the agent operates within an environment defined by an execution function $\mathcal{E}$. At each step $t_i$, the agent utilizes an observer, such as a text or a icon detector, to obtain the observation $o_i$ from the current environment state $s_i$. Subsequently, it runs $\mathcal{L}(q, \mathbf{M}, o_i)$ to predict the current action $a_i$. As the action $a_i$ is executed, the environment state transitions from $s_i$ to $s_{i+1}$ according to $\mathcal{E}(s_i, a_i)$.

This observe-act loop continues until the model predicts the stop action $a_i = \text{STOP}$ or reaches a predetermined termination condition, such as a maximum number of steps. The pipeline is illustrated in Figure 2(a). From this figure, it is evident that each action execution requires one perception of the environment from the observer. However, frequent reliance on the observer during long-horizon instructions could be time-consuming.

To address this challenge, our objective is to minimize the number of observations as much as possible in the execution step. The general pipeline can refer to the Figure 2(b). When observation cannot be bypassed, we also provide detailed prompts to facilitate action prediction. Leveraging the LLM's high accuracy with short instructions, we consider using short instructions to effectively resolve a long-horizon instruction. To achieve this, we first introduce a structural memory $\mathcal{SM}$ that encompasses various basic functions for each application on the PC, where each function consists of a sequence of instructions and actions, along with explanations for each action. Each basic function contains only 3-6 action steps. We then retrieve $\mathcal{SM}$ to select the appropriate basic function $\mathcal{F}$ for the instruction. We treat $\mathcal{F}$ as a foundational element and employ it to generate reusable subtask prompts and subsequent action prompts, thereby invoking the observer only when absolutely necessary. Next, we introduce two types of action-oriented fill-in-the-blank, subtask prompt and action prompt.

## 3.2 FILL-IN-THE-BLANK PROMPT

The ActionFiller framework consists of two types of action-oriented prompts: one designed for subtasks and another tailored for action sequences. In figure 3, we show the core mechanism behind this framework.

**Subtask Prompt:** The objective of the subtask prompt is to construct a coherent sequence of steps that reflects human experience while balancing effectiveness and operational flexibility. To achieve this, we introduce a Foresight Optimization Module (FOM). Initially, this module leverages past human experiences to generate a prompt that incorporates both reference steps—grounded in historical data—and additional operational steps that are adaptable to various contexts. Following this, a more detailed prompt is created, devoid of direct human experiential references, which outlines potential subtasks in a clear and organized manner. This second prompt is then optimized to address

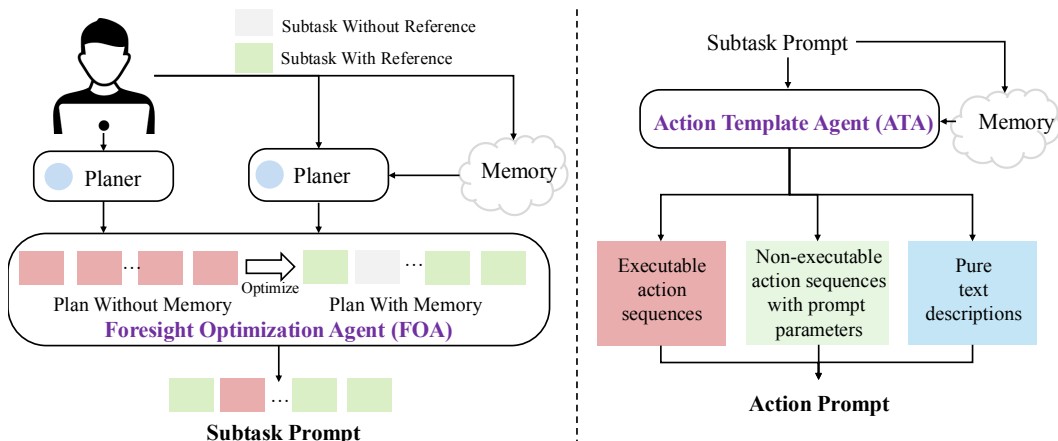

Figure 3: Fill-In-The-Blank prompt generation.

uncertain aspects by integrating both human experience and operational flexibility, ensuring that the agent can navigate complexities effectively. In Figure 4 (a), we provide a detailed demo to show how to generate subtask prompts.

**Action Prompt:** To emulate human thought processes in addressing complex problems, we have developed three types of action prompts to enhance familiarity with specific tasks. These prompts are designed to correspond to various levels of understanding, ranging from full mastery to initial recognition. By aligning the LLM's responses with the user's prior experience, the prompts help the agent deliver adaptive, context-aware actions, reducing redundant operations and improving overall efficiency.

In our paper, an Action Template Agent (ATA) utilizes the subtask prompt to generate three distinct types of action prompts for execution: 1) Executable Action Sequences: These templates are derived from human experiences and consist of tasks that involve short, clear instructions. This memory can be populated by the language model (LM) using predefined templates or through human input, tailored to specific contexts. Executable action sequences can be directly executed by the operating system agent, allowing for seamless interaction with the environment. 2) Unexecutable Action Sequences: These sequences are characterized by variable parameters, rendering them non-executable without additional context or information. Once the parameters are updated based on the current environmental state, they can transform into executable sequences, enabling the agent to adapt to changing conditions effectively. 3) Pure Textual Descriptions: This type emphasizes conveying actions through natural language, providing a narrative or illustrative format that is rich in detail. However, these descriptions often exceed the LM's immediate capabilities, necessitating further elaboration or contextual information for effective execution. This prompts the agent to seek additional input or clarification before proceeding. A detailed illustration is provided in Figure 4(b), where various types of action prompts are colored to distinguish between them.

### 3.3 COHESIVEFLOW AGENT

We observed that during action execution, the latter two types of prompts—non-executable action sequences with prompt parameters, and pure text descriptions—struggle to function effectively in the OS environment. To address this issue, we propose a CohesiveFlow agent, which focuses on either providing the correct parameters to render non-executable prompts executable, or optimizing textual descriptions to generate accurate, concise action sequences.

When encountering a second action prompt, we utilize a large language model (LLM) such as GPT-4 to predict the parameters for the next action based on previously executed actions and the current OS environment state. These parameters vary depending on the action type: for instance, a click operation requires coordinates, whereas a text-based action requires specific input. Rather than relying on traditional probabilistic models, this prediction task is framed as a sequence-to-sequence generation problem. The LLM predicts the next action $A_t$ from an input sequence that includes the users'

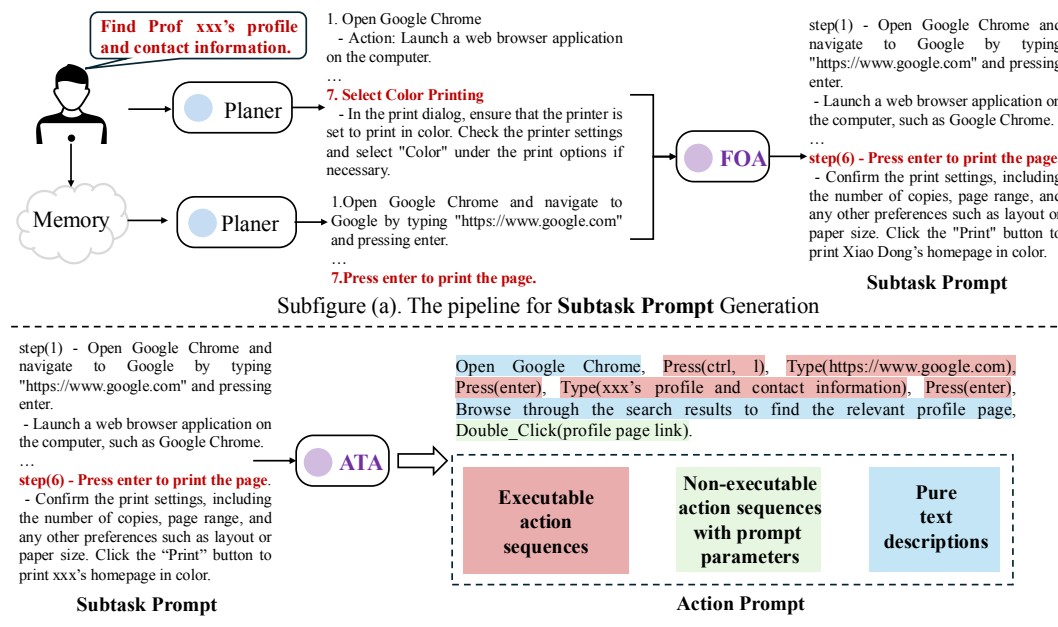

Subfigure (a). The pipeline for **Subtask Prompt** Generation

Subfigure (b). The pipeline for **Action Prompt** Generation

Figure 4: Demonstration of subtask and action prompt pipelines in OS agents

instruction $q$, current environment state $S_t$, past actions $A_{1:t-1}$, and possibly future actions $A_{t+1:}$ that may be executed. The prediction process is formulated as: $A_t = \mathcal{LLM}(q, S_t, A_{1:t-1}, A_{t+1:})$

When handling pure text descriptions $T$ that are non-executable, the LLM transforms the text into an optimized action sequence $A$ through the following process: $A = \mathcal{LLM}(q, T, S_t, A_{1:t-1})$

After executing the updated action at step $t$, we assess whether the action successfully achieves the intended goal. If the action fulfills its purpose, we adjust the remaining action sequence $\hat{A}_{t+1:}$ by considering the outcome of $A_t$. This update process is defined as: $\hat{A}_{t+1:} = \mathcal{LLM}(S_t, A_{1:t})$

Through iterative execution, our method leverages the LLM's capacity to infer complex relationships and dynamically adapt to the evolving state and context.

## 3.4 ENDUROSEQ DATASET

Figure 5: Dataset categories by solution path availability.

| Category | Task Examples | Tasks | % |
|---|---|---|---|
| Dynamic Tasks | Using Excel to plan weekly meals and check nutrition facts, then create a shopping list in Amazon to ensure all ingredients are available. | 15 | 50 |
| Static Tasks | Find rental apartments near NYU with a budget under $3,500 per month using Zillow, and compile the details (address, rent, and URL link) in a Microsoft Word document. | 15 | 50 |
| **Total** | | **30** | **100** |

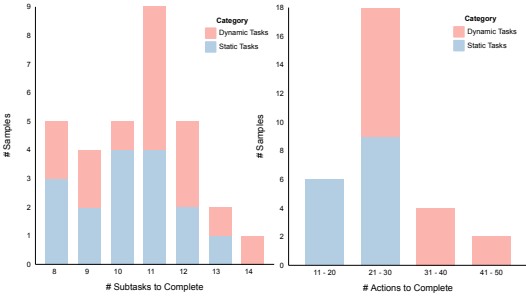

Figure 6: Characteristics and statistics of EN-DUROSEQ in subtasks and actions across samples.

To thoroughly evaluate the performance on long-horizon instructions, we introduce a novel dataset named EnduroSeq, specifically designed for this purpose. EnduroSeq consists of 30 carefully curated samples that are categorized into two distinct types of tasks: (1) Open tasks and (2) Static tasks. Open tasks are characterized by their flexibility, allowing for multiple possible solution paths

Table 1: Each sample in the ENDUROSEQ with a short task description.

| Category | Application | Type | Task Description |
|---|---|---|---|
| **Product Tools** | Chrome | Static | Change the settings of Chrome. |
| | | Static | Create bookmarks of several websites. |
| | Teams | Static | Schedule a meeting to corporate with team members. |
| | Google Docs | Static | Create and share a document in Google Docs. |
| | | Dynamic | Create a company report using Google Docs. |
| | | Dynamic | Create a project proposal using Google Docs. |
| | Office | Dynamic | Design and document a six-week HIIT bootcamp plan using Microsoft Word and Excel. |
| | | Dynamic | Write an outline for a speech at an international conference. |
| | | Dynamic | Develop a beginner gym strength training plan. |
| | Slack | Static | Create a channel in slack to corparate with teammates. |
| **Online Service** | Amazon | Static | Search and filter the items. |
| | | Static | Select a mattress that meets your requirements and buy it. |
| | Google Translate | Static | Translate a speech into French. |
| | Walmart | Static | Purchasing something in Walmart. |
| | Zillow | Static | Search apartments near NYU to rent. |
| | Coursera | Dynamic | Find and enroll a data science program. |
| | allrecipes.com | Dynamic | Organize a dinner party for six people by selecting recipes. |
| **Social Media** | Youtube | Static | Search and compile a list of quality Git learning tutorials. |
| | Spotify | Static | Create a playlist of your favorite songs. |
| | songkick.com | Dynamic | Check the play schedule of a band. |
| **Development Tools** | leetcode | Static | Find a java method of the "Two Sum" problem. |
| | jupyter notebook | Static | Use Windows cmd to create and configure a Jupyter Notebook file for machine learning. |
| | VSCode | Dynamic | Develop a Python web application using Visual Studio Code and Flask. |
| **Cross-App** | Web Browser, Office | Static | Search and organize a list of movies directed by Christopher Nolan on IMDb. |
| | | Static | Use Google to search for LA weather on weather.com and view it by month. |
| | Web Browser, Office | Dynamic | Find the Apple products information and compare two product. |
| | | Dynamic | Use Microsoft Office tools to do a personal wellness retreat. |
| | | Dynamic | Compile a comprehensive list of faculty members. |
| | Web Browser, Office, Amazon | Dynamic | Using Excel to plan Weekly Meals and check nutrition facts, then add them to Shopping List in Amazon. |
| | Office, Amazon | Dynamic | Design an outfit for the everyday man in spring on Amazon. |

to achieve the desired outcome. In contrast, static tasks are more rigid, offering only one predefined solution path that must be followed precisely. In Table 5, we present the breakdown of tasks categorized by solution path availability, showcasing examples, the number of tasks, and their respective percentages.

Each sample in the dataset is constructed to encompass a wide range of complexity, with over 8 sub-tasks and at least 11 action sequences, as illustrated in Figure 6. Each task and action sequence is designed to mimic real-world scenarios involving extended, multi-step instructions that challenge the model's ability to maintain context over long sequences. The detail of each instruction is shown in Table 1. Additionally, EnduroSeq is intended to facilitate the evaluation of various aspects of performance, such as adaptability to dynamic tasks and robustness in handling tasks with fixed constraints.

## 4 EXPERIMENTS

### 4.1 IMPLEMENTATIONS

In our experiments, the agent operates within a well-defined action space tailored to the Windows operating system. This action space consists of discrete actions, including basic navigation, selection, and interaction commands. Each action is mapped to the functional requirements of the environment, enabling the agent to efficiently progress through various tasks. For example, in the WindowsBench dataset, the action space includes task-specific interactions such as launching applications, navigating menus, and executing commands. The detailed action space is provided in Table 2.

| Action Name | Function Call | Description |
|---|---|---|
| Open app | `open_app('Teams')` | Opens the specified app, e.g., Teams. |
| Press | `press('Enter')` | Simulates pressing the 'Enter' key. |
| Type text | `type_text('amazon.com')` | Inputs a text string, e.g., 'amazon.com'. |
| Left click | `left_click(x, y)` | Performs a left click at coordinates (x, y). |
| Double click | `double_click(x, y)` | Double-clicks at coordinates (x, y). |
| Right click | `right_click(x, y)` | Right-clicks at coordinates (x, y). |
| Hover | `hover(x, y)` | Hovers over coordinates (x, y). |
| Swipe | `swipe(x1, y1, x2, y2)` | Swipes from (x1, y1) to (x2, y2). |
| Home | `home()` | Returns to the main interface. |

Table 2: Action space for agent interaction in our ActionFiller

Table 3: Performance comparison (%) on ENDUROSEQ.

| Framework | Category | SR | CR | Avg. SR | Avg. CR |
|---|---|---|---|---|---|
| `GPT-4o (Human Surrogate)` | Static Tasks | 40.0 | 62.3 | 56.7 | 73.8 |
| | Dynamic Tasks | 73.3 | 85.2 | | |
| `GPT-o1 (Human Surrogate)` | Static Tasks | 46.7 | 68.5 | 60.0 | 77.7 |
| | Dynamic Tasks | 73.3 | 86.8 | | |
| `ActionFiller` | Static Tasks | **80.0** | **91.8** | **80.0** | **92.7** |
| | Dynamic Tasks | **80.0** | **93.5** | | |

## 4.2 DATASET & BASELINES & METRICS

**Dataset** We utilize the **WindowsBench** and **ENDUROSEQ** datasets to evaluate the performance of our ActionFiller framework and baseline methods. WindowsBench, originally derived from the UFO agent, consists of 30 samples spanning 9 applications, along with a cross-application dataset containing a rich variety of operational samples. ENDUROSEQ is a custom dataset designed to include extensive subtasks and actions. It is segmented into static and dynamic categories, with the former targeting single-path solutions and the latter accommodating multi-path solutions.

**Baselines** In our paper, we use GPT-4o and GPT-o1 as our baseline.

**Metrics** We use two metrics to evaluate the performance of mobile device operation agents across different dimensions:

- **Success Rate (SR)**: This metric quantifies the agent's ability to successfully accomplish assigned tasks. A score of 1 is attributed when a task is fully completed, signifying successful execution.

- **Completion Rate (CR)**: This metric evaluates the agent's intermediate performance during task execution, specifically assessing the effectiveness of its actions. In scenarios requiring complex planning, even if the task is not fully completed, incremental progress or partially effective actions contribute positively to this score.

## 4.3 EXPERIMENT RESULTS

The experimental results in Table 3 highlight the superior performance of our proposed ActionFiller method across various applications in long instruction. For instance, in tasks such as Outlook and File Explorer, our method consistently achieves higher success rates and better Completion Rates (CR) compared to baseline approaches using gpt-4o and gpt-o1. Specifically, our method excels in handling complex applications like Visual Studio Code and Edge Browser, where it significantly outperforms the baseline, demonstrating its effectiveness in improving task completion and robustness. This strong performance across both static and dynamic tasks confirms the efficiency and reliability of the ActionFiller framework in optimizing agent actions and reducing errors.

To show the superiority of our ActionFiller in the short instruction, the experimental results in Table 4 show that ActionFiller significantly outperforms GPT-4 and GPT-o1 in the WindowsBench tests. Whether in terms of Success Rate (SR) or Completion Rate (CR), ActionFiller demonstrates

Table 4: Performance statistics for various applications in WindowsBench.

| Application | GPT-4 | | | GPT4-o1 | | | ActionFiller | | |
|---|---|---|---|---|---|---|---|---|---|
| | SR | Step | CR | SR | Step | CR | SR | Step | CR |
| Outlook | 100% | 8.4 | 73.9% | 60.0% | 7.6 | 76.2% | 100% | 6.5 | 96.0% |
| Photos | 40.0% | 7.0 | 32.7% | 60.0% | 6.8 | 35.7% | 80% | 3.2 | 93.7% |
| PowerPoint | 40.0% | 10.4 | 35.2% | 40.0% | 10.0 | 40.0% | 80% | 5.2 | 85.2% |
| Word | 20.0% | 9.2 | 15.3% | 40.0% | 8.4 | 40.0% | 80% | 5.4 | 81.5% |
| Adobe Acrobat | 0.0% | 7.6 | 40.2% | 0.0% | 6.4 | 42.9% | 40% | 4.7 | 75.6% |
| File Explorer | 80.0% | 6.2 | 63.4% | 80.0% | 9.0 | 72.7% | 100% | 4.8 | 88.7% |
| Visual Studio Code | 40.0% | 7.4 | 40.3% | 40.0% | 4.6 | 52.6% | 80% | 4.3 | 80.2% |
| WeChat | 40.0% | 6.2 | 68.0% | 40.0% | 6.4 | 72.0% | 80% | 5.6 | 83.1% |
| Edge Browser | 60.0% | 8.2 | 58.8% | 80.0% | 7.6 | 77.1% | 100% | 6.3 | 94.0% |
| Cross-Application | 0.0% | 13.8 | 49.7% | 0.0% | 13.4 | 60.6% | 60% | 10.4 | 73.5% |

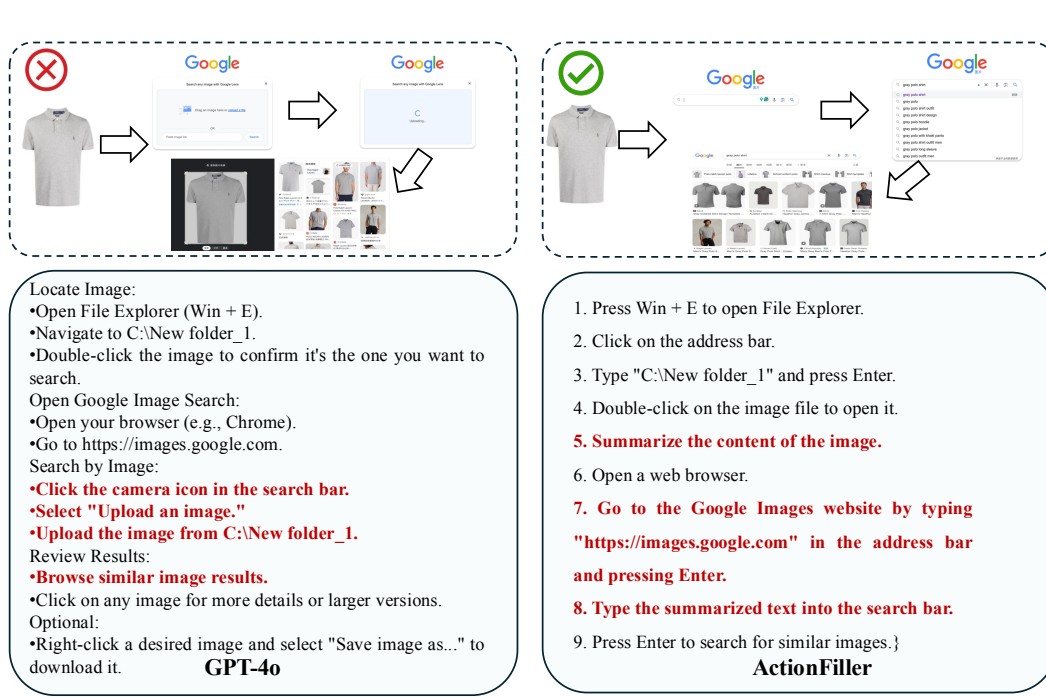

Figure 7: Comparison of generated plans between GPT-4 and ActionFiller.

higher efficiency and reliability across most applications, especially in key applications such as Outlook, Photos, and PowerPoint. It not only completes more tasks but also achieves a higher completion rate with fewer steps, showcasing its exceptional ability in handling complex tasks. This proves the strong advantages of ActionFiller in task automation within the Windows ecosystem.

## 4.4 CASE STUDY

In Figure 7, we present a demonstration using the instruction: 'Open the image at C: \\New folder_1, summarize its content, and use the summarized text to search for a similar image on Google.' The left side of the figure shows GPT-4's generated plan, while the right side displays ActionFiller's plan. We observe that GPT-4 misinterprets the intent of the instruction. In contrast, ActionFiller not only correctly comprehends the instruction but also provides a more efficient execution strategy, completing the task with a single action using the Observer (clicking the image). This further demonstrates the effectiveness and efficiency of ActionFiller.

We also show a successful episode in Figure 8, illustrating a successful episode where ActionFiller is employed to execute a long instruction for renting a house. The framework effectively decomposes the complex task into manageable subtasks, allowing for a structured and step-by-step approach.

Figure 8: Task Trajectory Using ActionFiller's Generated Plan in Long Instructions.

This ensures the agent remains adaptable to dynamic changes in the environment while maintaining a high level of accuracy and efficiency throughout the task.

## 5    CONCLUSION

In conclusion, this paper presents the ActionFiller framework as a novel approach to enhance the efficiency of operating system agents in executing long-horizon tasks. By leveraging the strengths of large language models and integrating short, executable tasks through innovative fill-in-the-blank prompts, ActionFiller minimizes redundancy and optimizes performance. The introduction of the Foresight Optimization Agent and Action Template Agent allows for the effective generation of action prompts, while the CohesiveFlow method ensures seamless execution by incorporating the current environmental state. Our experiments, validated by the EnduroSeq benchmark and conducted on the WinBench dataset, reveal significant improvements in task completion rates and execution efficiency. Overall, ActionFiller paves the way for more effective applications of intelligent agents in complex environments, demonstrating its potential to redefine task execution strategies in operating systems.

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
