# OpenReview forum: "ActionFiller: Fill-In-The-Blank Prompting for OS Agent"
_ICLR.cc/2025/Conference — Submitted to ICLR 2025_

### Official Review · Reviewer_Z7nS · 2024-10-31

**Soundness:** 2
**Presentation:** 3
**Contribution:** 2
**Rating:** 3
**Confidence:** 5

**Summary:**

This paper proposes the ActionFiller framework, an approach to improve task efficiency for operating system (OS) agents by reducing reliance on cognitive decision-making and creating action sequences through fill-in-the-blank prompts. Key contributions include the introduction of the Foresight Optimization Agent (FOA) and Action Template Agent (ATA), which generate task-specific action prompts, and the CohesiveFlow method, which optimizes unexecutable actions based on the environment's current state. Experiments on the EnduroSeq and WindowsBench datasets show improvements in task completion and execution efficiency.

**Strengths:**

- The paper presents a straightforward method for constructing LLM-based agents, which may appeal to those new to OS agents.
- The introduction of fill-in-the-blank prompts with action-oriented optimization methods attempts to address inefficiency issues in long-horizon tasks.
- Construct the EnduroSeq benchmark which is attempt to assess long-horizon instructions, which may benefit future research if further developed.

**Weaknesses:**

### No Novelty: The paper's methods remain basic, as they reflect standard LLM-agent construction. The fill-in-the-blank strategy and reliance on sequential action prompts lack depth or innovation compared to existing work.
- The core idea of using fill-in-the-blank prompting and breaking down tasks into subtasks is fairly standard in LLM applications
- The CohesiveFlow method appears to be a too straightforward application of masked language modeling without significant innovation
- Many components seem to be combinations of existing techniques rather than novel contributions
### Weak Experimental Support: The experiments are not sufficiently robust, and the evaluation metrics fail to convincingly validate the effectiveness of the proposed method.
- The exact mechanism of how parameters are updated in CohesiveFlow is not clearly explained
- No ablation studies to validate the contribution of individual components
- The evaluation is limited to only 30 samples in EnduroSeq, which seems insufficient for meaningful conclusions
- No statistical significance tests are provided for the results
- Limited comparison with state-of-the-art methods (only GPT-4o and GPT-o1 as baselines)
### Fundamental Contradiction in Efficiency Claims
- While the paper claims to improve efficiency for long-horizon tasks, the proposed method requires multiple LLM calls (for FOA, ATA, and CohesiveFlow), which inevitably increases the total computation time
- The paper only focuses on reducing the number of environment observations but ignores the significant computational overhead from multiple LLM interactions
- No runtime performance metrics or latency measurements are provided to support the efficiency claims
- This appears to be trading one form of inefficiency (environment observations) for another (increased LLM computation), without clear justification for why this is beneficial

**Questions:**

1. Can you provide more detailed justification for why 30 samples in EnduroSeq is sufficient for evaluation?
2. Can you provide ablation studies to show the contribution of each component?
3. How does the ActionFiller compare with more advanced state-of-the-art OS agent frameworks in terms of adaptability to real-world tasks?
4. Could additional datasets or benchmarks improve the framework’s applicability?
5. Can you provide comprehensive runtime measurements comparing your method against baselines, including the total time spent on LLM calls?
6. What is the average number of LLM calls required for a typical long-horizon task using ActionFiller? How does this compare to baseline methods?

---

### Official Review · Reviewer_VMWt · 2024-11-02

**Soundness:** 2
**Presentation:** 2
**Contribution:** 2
**Rating:** 3
**Confidence:** 3

**Summary:**

In this paper, the authors introduce *ActionFilter*, which is designed to integrate easily executable short tasks into longer, cohesive tasks using fill-in-the-blank prompt. The goal of the method is to mitigate the frequent reliance on observations for long-horizon instructions. The authors claim that the proposed method can minimize redundant operations to enhance overall efficiency of long-horizon os operation tasks. Experiments are conducted on the long horizon os benchmarks the authors designed, called *EnduroSeq*, where ActionFilter shows superior performance compared to GPT-4o and GPT-o1 baselines.

**Strengths:**

- Minimizing the reliance on observations for foundation model agents is an interesting topic.
- The authors show some interesting results of OS agent solving long horizon tasks.
- The authors propose to keep the planning action sequence in 3 formats: executable actions, non-executable ones requiring parameters, and pure text descriptions and only translate them to actions at execution, making the high level planning flexible.

**Weaknesses:**

- The experiments are designed in a confusing way
  -  Unfair comparison to the baselines and missing more relevant baselines:
     -  Is the structural memory SM actually accessible to the 4o and o1 agents? While ActionFilter has the access to the basic functions, it is intuitive that it will perform better. However, the authors don't compare with other agents that can utilize such resources, such as using it as the skill library in Voyager[1], the state-aware guidelines in AutoGuide[2], or an adapted version of RAG like RAP[3].
     - What is the language model that the authors use for ActionFilter? In L265-266 it says the method uses GPT-4 to predict parameters, what about other modules? Why do the authors compare to GPT-4o and o1 as the baselines? In that case shouldn't the results for ActionFilter(4o) and ActionFilter(o1) reported separately?
  -  Experiments are only conducted in one benchmark, its ability in some more commonly used benchmarks like VisualWebArena[4] is unclear.
  - The authors aim to reduce the reliance on getting observations. However, there lack experiments of how many observations are used by the method and the baselines.
- Limited adaptivity and generalization ability of the method to other domains
  - The authors assume that executable action sequences are directly grounded to the environments, therefore the action generation module can directly applying it without retrieving and processing observations. However, this seems to be a very extreme condition where nothing unexpected should happen in the environment, which is unrealistic for most domains. Sometimes even there are some changes in the environment, the action can still be executable, but lead to unexpected next step, in this case the algorithm will be fooled by the environment.

**Questions:**

- Where is the Reflector in Figure 1 and 2 used in the process?
- What is the size of the evaluation dataset?
- In L298-299, how does the agent assess whether the action successfully achieves the intended goal or not?
- There are open tasks and dynamic tasks. What are the differences? From the explanations in the paper they seem to be the same type of tasks.

---
[1] Wang, G., Xie, Y., Jiang, Y., Mandlekar, A., Xiao, C., Zhu, Y., ... & Anandkumar, A. (2023). Voyager: An open-ended embodied agent with large language models. arXiv preprint arXiv:2305.16291.

[2] Fu, Y., Kim, D. K., Kim, J., Sohn, S., Logeswaran, L., Bae, K., & Lee, H. (2024). Autoguide: Automated generation and selection of state-aware guidelines for large language model agents. arXiv preprint arXiv:2403.08978.

[3] Kagaya, T., Yuan, T. J., Lou, Y., Karlekar, J., Pranata, S., Kinose, A., ... & You, Y. (2024). Rap: Retrieval-augmented planning with contextual memory for multimodal llm agents. arXiv preprint arXiv:2402.03610.

[4] Koh, J. Y., Lo, R., Jang, L., Duvvur, V., Lim, M. C., Huang, P. Y., ... & Fried, D. (2024). Visualwebarena: Evaluating multimodal agents on realistic visual web tasks. arXiv preprint arXiv:2401.13649.

---

### Official Review · Reviewer_agmm · 2024-11-03

**Soundness:** 2
**Presentation:** 2
**Contribution:** 2
**Rating:** 3
**Confidence:** 4

**Summary:**

This paper proposes the ActionFiller framework, which uses a fill-in-the-blank prompting method to enhance the efficiency of existing OS agents when cognition modules are overused in long-horizon tasks. The framework divides tasks into subtasks through a Foresight Optimization Agent (FOA) and generates action sequences using an Action Template Agent (ATA). Additionally, a new dataset called EnduroSeq is introduced to evaluate OS agent performance.

**Strengths:**

- The paper introduces a method to address inefficiencies caused by excessive reliance on cognition modules in existing OS agents.
- Introducing a dataset specifically designed to evaluate OS agent performance is a valuable contribution.

**Weaknesses:**

- Many key elements of ActionFiller are insufficiently explained. For example:
    - In Figure 2, the roles of the Planner, Observer, Actor, and Reflector need clarification. There should be a loop between action generation and observation in Figure 2(a) that is not depicted. Additionally, the Reflector appears in the figure without prior mention.
    - The process for generating Subtask Prompt and Action Prompt, which is the most critical components, is insufficiently explained. For example, the meaning and role of reference steps and operational steps in subtask prompt generation are not defined. It is also unclear what inputs the planner receives and how it generates its two outputs. Further explanation is needed on how the ATA distinguishes and generates the three action classes in the Action prompt.
    - Memory usage is unclear. The process for building and using memory is not fully explained. If predictions are made using the entire memory $M$ with $L(q, M, o_i)$, this could be confusing. If specific examples are retrieved, the retrieval mechanism (e.g., similarity measure, number of examples, memory construction method) should be explained. This also applies to Structural Memory ($SM$).
    - Both the input and output of the LLM are referred to as a prompt, making ActionFiller’s operation difficult to understand.
- Although the paper highlights efficiency issues in existing OS agents, the proposed method also appears to have limitations. Both FOA and ATA require multiple LLM calls. Additionally, in the CohesiveFlow Agent, $A_t = LLM(q, S_t, A_{1}, A_{t+1:})$ is used to generate $A_t$, and $\hat{A_{t+1:}} = LLM(S_t, A_{1})$ after execution. The increased input token usage suggests that this method may not fully resolve the efficiency issue.
- The contributions of the proposed EnduroSeq dataset need emphasis. It would be helpful to clarify its strengths compared to WinBench, and address concerns about its limited sample size of 30.
- The experiments section does not compare against UFO as a baseline.
- Including experiments that directly measure ActionFiller’s efficiency would be beneficial. To validate the claimed efficiency improvements, data on the number of input tokens or execution time used in subtask and action prompts should be provided.
- Minor The citation style needs correction.
- Minor The conclusion should address limitations and suggest directions for future work.
- Appendix Including the actual prompts used and ActionFiller’s outputs in the appendix would aid understanding.

**Questions:**

- Why was UFO not included as a baseline in the experiments?
- Could you explain how the EnduroSeq dataset presents more challenges compared to WinBench?
- Is it possible to directly compare efficiency with baselines? For example, can the total input tokens used or execution time for ActionFiller be compared with those of other baselines?

---

### Official Review · Reviewer_RQsf · 2024-11-04

**Soundness:** 2
**Presentation:** 1
**Contribution:** 2
**Rating:** 3
**Confidence:** 4

**Summary:**

The paper introduces ActionFiller to enable task automation through OS agents by using fill-in-the-blank prompting to tackle long-horizon tasks with multiple application interactions. Two hierarchical modules, FOA and ATA, generate task flows in action sequences by leveraging past experiences and generating actionable subtasks, reducing redundant steps. Experiments demonstrate that ActionFiller improves completion rates over the baselines with GPT-4o and GPT-o1, validated by the EnduroSeq benchmark.

**Strengths:**

- By leveraging LLMs to interpret and convert user natural language commands into feasible actions, the ActionFiller framework demonstrates practical potential for complex task automation, aiming to enhance efficiency in task and action sequence planning through experience- and template-based prompting.

**Weaknesses:**

- The novelty and contributions of this work are not clearly distinguished when compared to recent advancements in LLM-based mobile task automation such as

[1] MobileGPT: Augmenting LLM with Human-like App Memory for Mobile Task Automation ( https://arxiv.org/html/2312.03003v3 )

[2] AutoDroid: LLM-powered Task Automation in Android ( https://arxiv.org/abs/2308.15272 )

[3] Mobile-Agent: Autonomous Multi-Modal Mobile Device Agent with Visual Perception ( https://arxiv.org/abs/2401.16158 )


- The proposed ActionFiller framework is compared only with GPT-4 and GPT-o1 in experiments. Extending comparisons to include these recent works utilizing advanced LLMs for mobile task automation could better highlight the novelty and contributions of ActionFiller.

- Also, there should be descriptions on how these pure GPT-4 and GPT-o1 baselines were implemented for comparison.

- The main solution of this paper is to perform observations only when necessary. However, there are no experimental results showing that the number of observations has been reduced according to this problem definition.

- The framework relies on template-based prompting, which may limits its adaptability to tasks that deviate significantly from predefined structures, potentially reducing flexibility in handling novel commands.

- The framework may encounter scalability issues as the memory of past actions grows, potentially slowing down the system as it searches for relevant past experiences and templates.

- ActionFiller’s use of LLMs to interpret and dynamically generate prompts can be computationally expensive, which might restrict its efficiency on mobile devices with limited processing power.

- The paper lacks sufficient implementation details on which LLM is used for ActionFiller, how prompts are designed for FOA and ATA, how current observation is represented, and how the memory structure is implemented to effectively retrieve relevant functions in response to commands.

- Figure 3 appears unnecessary, as it largely overlaps with Figure 4. A unified diagram encompassing FOA, ATA, Memory, and the Cohesiveflow Agent, illustrating the structure and operation of each component, would improve comprehension of the paper. Additionally, most figures present only abstract information and should include more specific details. For instance, in Figure 1, the distinction between the left and right sides and their impact on the OS environment should be clarified.

- The ablation study appears insufficient. There should be an investigation into the use of Memory both with and without FOA, as well as an analysis of Memory size relative to the number of tasks. Additionally, while the concept of an ActionFilter auto-pipeline is promising, it seems to rely heavily on the performance of LLMs. Therefore, an analysis of ActionFilter performance across various LLMs, starting with smaller models, is necessary to evaluate the impact of LLM size on performance.

- Typo: Lines 41~48, please add space or parentheses for citation.

**Questions:**

See the weaknesses.

---

### Meta-Review · Area_Chair_AMKY · 2024-12-18

**Metareview:**

The paper presents an approach for OS agents focused on long horizon tasks. While the motivation of this work is exciting, the reviewers have raised several concerns on novelty and execution. The authors have not responded to the reviewer's comments and hence not addressed their concerns. For the next version of this paper, I would recommend a restructure of the paper with an emphasis on the novel aspects of the work. On the experimental side, I would recommend looking at conducting exhaustive ablation analysis of your method along with experimenting with more comprehensive benchmarks.

**Additional Comments On Reviewer Discussion:**

The authors did not respond to the reviewers. Hence there was no discussion on this paper.

---

### Decision · Program_Chairs · 2025-01-22

Reject